Differential contribution of soil biota groups to plant litter decomposition as mediated by soil use

Castro-Huerta Ricardo A. ricardoach@yahoo.com
Falco Liliana B.
Sandler Rosana V.
Coviella Carlos E.
Ecology Laboratory, Terrestrial Ecology Research Program, Basic Sciences Department—Ecology and Sustainable Development Institute, National University of Luján , Luján, Buenos Aires , Argentina
Ritsema Coen
Electronic publication date: 2015 Mar 5
Publication date: 2015
Volume: 3
Electronic Location ID: e826
Received 2014 Dec 18; Accepted 2015 Feb 17
Copyright: © 2015 Castro-Huerta et al.
Copyright year: 2015
Copyright holder: Castro-Huerta et al.
License: This is an open access article distributed under the terms of the Creative Commons Attribution License, which permits unrestricted use, distribution, reproduction and adaptation in any medium and for any purpose provided that it is properly attributed. For attribution, the original author(s), title, publication source (PeerJ) and either DOI or URL of the article must be cited.
License URL: https://creativecommons.org/licenses/by/4.0/

Keywords: Litterbags, Organic matter turnover, Soil use, Soil fauna, Agroecosystem, Soil sustainability, Litter decomposition, Edaphic biota, Nutrient cycling, Agricultural intensity

Funding: Ministry of Science and Technology of Argentina Universidad Nacional de Luján This work was partially funded by a grant from the Ministry of Science and Technology of Argentina, Project PRH 19, and by the Universidad Nacional de Luján. The funders had no role in study design, data collection and analysis, decision to publish, or preparation of the manuscript.

==============================
Plant decomposition is dependant on the activity of the soil biota and its interactions with climate, soil properties, and plant residue inputs. This work assessed the roles of different groups of the soil biota on litter decomposition, and the way they are modulated by soil use. Litterbags of different mesh sizes for the selective exclusion of soil fauna by size (macro, meso, and microfauna) were filled with standardized dried leaves and placed on the same soil under different use intensities: naturalized grasslands, recent agriculture, and intensive agriculture fields. During five months, litterbags of each mesh size were collected once a month per system with five replicates. The remaining mass was measured and decomposition rates calculated. Differences were found for the different biota groups, and they were dependant on soil use. Within systems, the results show that in the naturalized grasslands, the macrofauna had the highest contribution to decomposition. In the recent agricultural system it was the combined activity of the macro- and mesofauna, and in the intensive agricultural use it was the mesofauna activity. These results underscore the relative importance and activity of the different groups of the edaphic biota and the effects of different soil uses on soil biota activity.

Introduction

The edaphic biota is the main factor directly responsible for soil organic matter turnover and nutrient cycling due to the diversity of processes in which it takes part (Lavelle & Spain, 2001; Lavelle et al., 2006; Brussaard, de Ruiter & Brown, 2007; Culliney, 2013). Among these, the fragmentation and incorporation of plant residues into the soil; the construction and maintenance of the structural porosity and soil aggregation are some of the processes the edaphic biota is involved with that have effects on other organisms (Lavelle et al., 2006; Culliney, 2013). This way, multiple interactions with other organisms are developed, on different scales and through the entire range of chemical, physical, and biological processes that support the ecosystem services provided by the soil (Brussaard, de Ruiter & Brown, 2007; Culliney, 2013). Lavelle et al. (1993) found that the activity of macroorganisms is particularly important in the regulation of the decomposition process. In particular, these authors state that interactions between macro- and microorganisms are very intense in areas where climate is more or less constant. In turn, this biological interaction is associated with the energy source, fungi and bacteria communities, and macroorganisms which create conditions suitable for optimal microbial activity. When environmental traits like the climate are not limiting (drought or flooding) and clay minerals are not very reactive (or do not make a significant contact with the biota), the biological regulation systems take a predominant role in the decomposition of leaf litter (Lavelle et al., 1993; Gonzalez & Seastedt, 2001; Dechaine et al., 2005).

Soil fauna largely control the decomposition process through breakdown of litter. digestion, and stimulation of microorganism activities (Yang & Chen, 2009). Their study indicated that soil fauna assemblage provided a significant contribution to litter decomposition in all three sites of their study (rainforest, broad-leaf forest and secondary forest), while the contribution of soil fauna to plant litter decomposition was more pronounced in the rainforest than the other two sites. Fauna effects increased N concentration and decreased C concentration in litter with high initial C/N ratio, which may explain the significant fauna effect on litter decomposition in the rain forest. A similar observation regarding the relation C/N was presented by Li et al. (2014).

The edaphic biota is classified according to the size of the adults into three groups: the microfauna, the mesofauna, and the macrofauna (Lavelle & Spain, 2001; Eisenbeis, 2006; Lavelle et al., 2006). Each component fulfills a specific role in its specific ecological niche that is hard to replace with other components present in the system (Lavelle et al., 2006) taking part in different processes affecting soil fertility in at least two main ways. Firstly, by promoting decomposition directly through the conversion of plant litter into their own tissues and indirectly transforming the plant litter physically and chemically into substances amenable to further degradation by microflora. Secondly, by their effects on the physical structure of the soil (Culliney, 2013) that may be affected by agricultural soil use (Baker, 1998; Bardgett & Cook, 1998)

Agroecosystems are continuously under the anthropic impact of different agricultural practices, which causes changes in their biotic and abiotic components both in time and in space. These changes in turn, affect the structure and function of the soil biota (Domínguez et al., 2014; Liiri et al., 2012), thus mediating the biological processes in the soil, which affects the flow of matter and energy in the entire system (Lubchenko et al., 1991). Ponge et al. (2013) show that soil animals (except epigeal springtails) and microbial community are adversely affected by the increase in agricultural intensification. The soil fauna responds to the agricultural management as a result of the physico-chemical disturbances that are produced in its habitat, of the distribution of the residues, and of the plant communities present (Lavelle & Spain, 2001; Kautz, Lopez-Fando & Ellmer, 2006).

In order to understand in greater detail the role of the different soil fauna groups in the decomposition process, the hierarchical model proposed by Lavelle et al. (1993) was followed. To isolate the effects of different agricultural management practices, fields with the same soil and climate in three levels of agricultural use intensity were selected, and factors such as resource quality were standardized. To assess the different contributions of the soil fauna in the decomposition process the technique of the litterbags (Crossley & Hoglund, 1962) was used.

Three litterbag mesh sizes were used to hierarchically exclude each group of the edaphic biota according to size. Therefore, the contribution of each group to organic matter decomposition was evaluated.

The working hypothesis was that plant litter decomposition rates would differ between the different soil fauna groups, and that those differences would be modulated by the different soil use intensities.

Materials and Methods

The study was carried out in the rolling pampas of central Argentina. With over fifty million hectares of agricultural land, it is one of the biggest and most productive plains in the world (Navarrete et al., 2007; Faggi et al., 2008). Three agroecosystem types with different intensities of soil use were selected as treatments. The agroecosystems were located near Chivilcoy city in the Buenos Aires province, Argentina (35°03′00″S; 59°41′00″W) (Fig. 1).

Figure 1 Sampling location.

Map showing the location of the sampling sites.

The soil for all treatments was a mollisol from the typical arguidoll group (Soil Survey Staff, 2010). In increasing order of soil use intensity, the selected agroecosystems were: 1-Naturalized grasslands with no anthropic impact in almost 50 years (NG). Natural litter in this agroecosystem was typically from naturally occurring grasses dominated by Festuca sp., 2-Cattle-grazing fields turned into agriculture 2 years before the start of the study (RA). Litter in this place was dominated by agricultural species, such as soybean and corn in summer and wheat in winter, and 3-Intensive Agriculture fields with almost 40 years of continuous and intensive agriculture (IA). Litter in this place was dominated by soybean. All the fields were left fallow during the duration of this experiment. The experimental sites ranged in size from 0.5 to 45 hectares.

Five different sites per treatment (soil use intensity) were selected as replicates. At each site, decomposition bags (20 × 20 cm) were placed with three different mesh sizes for the selective exclusion of the epigeic soil organisms according to size: 4 mm mesh size (Microfauna + Mesofauna + Macrofauna, hereafter Total Biota); 2 mm mesh size for the selective exclusion of the Macrofauna (Microfauna + Mesofauna); and 0.25 mm mesh size to further exclude the Macrofauna and Mesofauna (hereafter Microfauna). These different size-excluded groups represent three different complexity groups of the soil biota.

In each bag 5 g of dry, senescent soybean leaves (Glycine max L.) were placed. Soybean leaves were used to standardize the litter material offered, because it was the last crop in the agricultural systems, and it has been the most common crop in the region during the last ten years. The senescent leaves were collected in the same field on autumn before harvest and dried at 30 °C. The bags were placed on the surface of the soil and covered lightly with plant residues after harvest to improve the natural decomposition of this crop in the pampas on end-autumn, winter and spring. A total of 270 litterbags were distributed among treatments and replicates, with no more than 10 bags per square meter at any site. The experiment was set to run for six sampling dates, but it was terminated after five months when one of the replicates approached zero RM. Bags were retrieved at 17, 53, 94, 126, and 171 days after the bags were placed. Every sampling date, one bag of each mesh size (3) was retrieved per replicate (5) and agroecosystem (3), thus processing 45 litterbags each sampling date, for a total of 225 litterbags over the 5 sampling dates. The material was then dried at 30 °C to constant weight. The remaining material was weighted and the percentage of remaining mass (%RM) calculated.

With these data, we performed a two-way ANOVA for discriminate the effects of both factors mesh-size and agricultural use, in case of found any difference, the decomposition rate for each case was calculated, assuming a negative exponential model following Olson (1963): RM=IM∗e−kt

where: RM=Remaining Mass

IM=Initial Mass

t=Time (Days)

k=Decomposition rate.

This exponential model was linearized using the natural logarithm of the remaining mass (%RM) and the transformed data were analyzed with ANCOVA to compare the slopes of each case with the Tukey test (HSD) between different cases with a 95% confidence interval.

All the statistical analysis were performed under the R Development Core Team (2010).

Results

The results of percentage remaining mass (%RM) found for each edaphic biota group within agricultural systems analyzed by two-way ANOVA can be seen on Table 1. Statistical differences were found for the mesh size factor. No differences were found for agricultural use alone, or the interaction between main factors.

Table 1 ANOVA table.

Two-way ANOVA for Mesh size, Agricultural use and the interaction. No significant interaction or effect of agricultural use, mesh size is the main significant effect found.

Two-way ANOVA	SS	DF	MS	F	p-value	
Intercept	4,758.675	1	4,758.675	19,980.80	0.000000	
Agricultural use	0.682	2	0.341	1.43	0.240816	
Mesh-size	6.427	2	3.214	13.49	0.000003	
Agr-use * Mesh-size	0.685	4	0.171	0.72	0.579533	
Error	62.160	261	0.238			

These results allow for the analysis of the slopes of the decomposition rates between different mesh sizes within agricultural uses (Figs 2–4). A covariance analysis using the natural logarithm values of the remaining mass (in %RM) measured for the different groups of the soil fauna showed statistically significant differences for the different groups within each soil use (Tukey HSD, p < 0.05).

Figure 2 Remaining mass for the Naturalized Grassland system.

Results for remaining mass (%) found for the Naturalized Grassland. A significant reduction in decomposition rate occurs when the Macrofauna is excluded. Negative exponential curve and R2 values are shown for each fauna group. Decomposition rate (k) corresponds to the loss of mass per day. Different letters indicate significant differences (In MR%) through covariance analysis contrasted with Tukey HSD test (α < 0.05).

Figure 3 Remaining mass for the Recent Agriculture system.

Results for remaining mass (%) found for the Recent Agriculture. Significant reduction in decomposition rate occurs when both Macro- and Mesofauna are excluded together. Data shown as in Fig. 2.

Figure 4 Remaining mass for the Intensive Agriculture system.

Results for remaining mass (%) found for the Intensive Agriculture. Significant decrease in decomposition rate occurs when the Mesofauna is excluded. Data shown as in Fig. 2.

In the less disturbed system, the Naturalized Grassland (Fig. 2), the decomposition rate (k) calculated for the Total Biota was significantly higher and different (p < 0.05) from the other two groups, with no differences between them. In the Recent Agriculture system (Fig. 3), the decomposition rate (k) of the Total Biota was significantly higher than that of the Microfauna alone, while the Microfauna + Mesofauna group did not differ with the other two groups. In the Intensive Agriculture system, the decomposition rate of the Microfauna was significantly lower than the other two groups that did not differ from each other (Fig. 4).

When analyzed between systems, the only significant difference found in decomposition rate was for the Microfauna. Decomposition rate for this group was significantly higher in the less disturbed system (NG) when compared to the two agricultural systems (Fig. 5).

Figure 5 Remaining mass results for the Microfauna.

Results for remaining mass (%) found for the Microfauna when compared across treatments; Decomposition rate due to Microfauna activity is higher in the less anthropized system when compared to both agricultural ones. No significant differences were found for the Macro- or the Mesofauna across systems. Data shown as in Fig. 2.

Overall, a total of five collembola families, twenty-one acari superfamilies and nine earthworm species were identified in all the sampling sites altogether (data not shown).

Discussion

In this work, two of the selected agroecosystems had been each under the same use for several decades, while the third system was of intermediate disturbance. It was assumed that by the time of the experiment, the structure and composition of the edaphic fauna in each system was already in equilibrium with the inputs and local use regime of each agroecosystem. Therefore, the results on decomposition rates are proper of the fauna already adapted to each agroecosystem.

In this way, the original edaphic fauna of the region was assumed to be best represented by the edaphic fauna in the NG system. In the Intensive Agriculture system, the original fauna was subjected to decades of strong habitat changes brought about by tillage, fertilizers, pesticides, changes in temperature regime, water availability, and soil structure (Domínguez et al., 2014).

The results show that the relative contribution of each fauna group to decomposition rate is different between systems, which is an indication that the structure and composition of the edaphic fauna is different from one system to another.

In the Naturalized Grasslands, the less anthropized system, the only significant difference was found when the Macrofauna was excluded. This is an indication that it was this group the one that contributed most significantly to decomposition in this system (k = 0.0074), since no difference was found when the Mesofauna was further excluded (k = 0.0032) (Fig. 2). In the system with intermediate anthropic impact, Recent Agriculture, the only statistical difference occurred when comparing Total Biota (k = 0.0088) with Microfauna alone (k = 0.001) (Fig. 3). This result indicates that the most significant contribution to decomposition in this system is the interaction of the Macrofauna with the Mesofauna together. Indeed, no difference was found when only the Macrofauna or only the Mesofauna were excluded. In the Intensive Agriculture system, the group that significantly contributed to the decomposition was the Mesofauna because the only significant difference in this agroecosystem occurred with its exclusion that lowers the decomposition rate (k) from 0.0086 to 0.0025 (Fig. 4).

When examined across systems, only the Microfauna presented significant differences between agroecosystems (Fig. 5). This group showed significantly higher decomposition rate in the less impacted system when compared to both intermediate and high anthropic impacts. No significant differences were found between systems for either Total Fauna or Mesofauna + Microfauna groups. This result underscores the sensitivity of the Microfauna to anthropic activities.

Interestingly, and despite what other authors found for similar soils (Domínguez et al., 2014), there were no significant differences due to land use intensity alone or its interaction with the edaphic biota. This is consistent with Geissen, Peña-Peña & Huerta (2009), who found that intensive management in Banana fields in Mexico did not differ in litter decomposition rate when compared to successional forests on the same soil.

In interpreting the results presented, it is important to keep in mind that the litterbag system employed, and the way they were deployed on the soil surface, attracts mainly epigeic soil fauna. Therefore, most of the results are relevant mainly to this group.

The results obtained in this work clearly support the working hypothesis. The data show that the different groups of the epigeic soil biota contribute differently to litter decomposition, and that this differential contribution is being mediated by the differences in soil use. Interestingly enough, these groups present complementary activities depending on soil use intensity, since no differences in decomposition rates were found between uses when the decomposition rates of the entire soil biota were compared. However, the contribution of each group of the soil fauna to the total litter decomposition changed across soil uses.

Being the less disturbed system, the Naturalized Grassland is the closest to the original, pristine condition of the soil systems studied. In these conditions, the Macrofauna is the relevant group as reflected by the lowering of the decomposition rate brought about by its selective exclusion with no further effect when the Mesofauna was also excluded. Indeed, results from previous authors (Brussaard, de Ruiter & Brown, 2007; Kampichler & Bruckner, 2009) indicate that in the less disturbed ecosystems, the contribution of the Mesofauna to litter decomposition is only marginal respect to that of the Macrofauna. Those results are in agreement with the ones presented in this work.

In the Recent Agriculture system, which is of intermediate disturbance, the combined effects of both the Macrofauna and Mesofauna are relevant, since the only significant difference is the result of their exclusion together.

The results from the most disturbed system, the Intensive Agriculture, show the Mesofauna as the relevant group, for the only significant decrease in decomposition rate takes place when this group is excluded. These results are thoroughly consistent with the hierarchical model proposed by Lavelle et al. (1993).

When analyzed across systems the results for the Microfauna showed higher decomposition rates for the least anthropized Naturalized Grasslands when compared to both agricultural systems. No differences were found for the Total Fauna or the Macro + Mesofauna across systems. These results suggests a negative effect of the agricultural practices on the Microfauna in particular. This is consistent with previous research showing evidence that agricultural management affects the structure of the microbial community (Wakelin et al., 2009; Zhong et al., 2010). Results by Lavelle et al. (2006). It also indicates that this particular group is very sensitive to any disturbance of the soil environment.

Despite their recognized importance, the interactions between the different groups of the soil biota are still largely unknown and one of the most studied topics in soil ecology (Hättenschwiler, Tiunov & Scheu, 2005; Fitter et al., 2005; Kampichler & Bruckner, 2009; Culliney, 2013). Coûteaux et al. (1991) and Bradford et al. (2002) found a significant increase in the decomposition rates when these three groups (micro-, meso- and macrofauna) were found acting together, when compared to less complex soil fauna groups.

It was also found that under organic agricultural management, the Mesofauna increases its abundance (Kautz, Lopez-Fando & Ellmer, 2006; Peredo et al., 2009) possibly due to the specialization in the consumption of the sources of litter left by crops (Kautz, Lopez-Fando & Ellmer, 2006; Milcu et al., 2006). The results presented here also support that assumption.

In terrestrial ecosystems, the empirical evidence is scarce but it is known that when a soil community has a high diversity of functional traits, it has effects that facilitate interactions promoting decomposition (Gessner et al., 2010). More evidence from size-exclusion studies is needed in order to thoroughly assess to what extent agricultural practices affect soil fauna diversity (Hättenschwiler, Tiunov & Scheu, 2005; Gessner et al., 2010; Culliney, 2013) and to improve agricultural practices for soil biota conservation that ensure decomposition and mineralization processes in agroecosystems.

The continuous disturbances in the studied agricultural systems could be selectively pressing certain organisms or groups over others. In this way, disturbances would allow for the establishment and development of soil biota adapted to anthropized systems, in detriment of the original soil biota, most likely close to the one in the Naturalized Grassland system.

From the results presented here, it can also be concluded that Microfauna is the most sensitive group to anthropic disturbances, and therefore, it should be the group to be taken particularly into account when devising sustainable agricultural practices.

Conclusions

In conclusion, when looked at them separately the different soil uses studied in this work strongly modulate the decomposition activity of each group of the soil fauna, even though the total decomposition rate remains the same for all the studied systems when the whole soil biota is present. These differences are likely due to changes in the structure and functioning of each one of the faunal groups of the soil biota studied, brought about by the different soil use intensities.

Finally, the results shown in this work point to a replacement of the relative contribution to decomposition of the different faunal groups as use intensity increases. In the less disturbed environments, it is the Macrofauna group which contributed the most to decomposition; this is consistent with the hierarchical model presented by Lavelle et al. (1993). As intensity of use increases, the Mesofauna activity gains in relative importance, being the most important group in the most disturbed environment.

Supplemental Information

Supplemental Information 1 Castro-Huerta et al. raw data

Remaining mass for all treatments and sampling dates

Click here for additional data file.

The authors wish to thank Edgardo Ferrari and Pablo Peretto for allowing the use of their properties as sampling sites. Loreta Gimenez provided great help with the fieldwork and laboratory analyses. The comments by Dr. Esperanza Huerta and two other anonymous reviewers helped to greatly improve the final draft of this manuscript.

Additional Information and Declarations

Competing Interests

Author Contributions

The authors declare there are no competing interests.

Ricardo A. Castro-Huerta conceived and designed the experiments, performed the experiments, analyzed the data, wrote the paper, prepared figures and/or tables, reviewed drafts of the paper, field work, making the litterbags.

Liliana B. Falco conceived and designed the experiments, performed the experiments, analyzed the data, wrote the paper, prepared figures and/or tables, reviewed drafts of the paper, field work.

Rosana V. Sandler and Carlos E. Coviella conceived and designed the experiments, analyzed the data, wrote the paper, reviewed drafts of the paper, field work.

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
