# Peer review of "Differential contribution of soil biota groups to plant litter decomposition as mediated by soil use"

_PeerJ, doi:10.7717/peerj.826_

## Round 0.1 · original submission · Minor Revisions

Overall, the indicated remarks and suggestions of all 3 reviewers should be addressed accordingly by the authors to improve the manuscript and make it acceptable for publication in PeerJ

·

Basic reporting

No comments

Experimental design

I suggest to add a figure with the location of the replicas and the position of the litter bags per meter square. Please indicate the number of bags per treatment, per measure time and per replica.

Validity of the findings

Yes the conclusions are relevant, but it is important to know the quality of the litter in each land use, in order to discuss more why was more relevant the presence of one group or another. Also if possible it is important to couted previous soil edaphic works, for knowing which epigeic species are abundant in each group.

Additional comments

It is a very interesting work, but some informations must be added, please in itroduction try to cout also works from 2010-2014, please talk more about the quality of litter and its relationship with soil fauna, in a succesional perspective, and according to each land use. It is important to emphasize that your sampling method enhance the capture of epigeic organisms, and that is important also to mention in your discussion.

Reviewer 2 ·

Basic reporting

The expression of numeric data in figures and table is not correct, such as p-value in table 1 should be 0.240816 instead of 0,240816.

Experimental design

No Comments

Validity of the findings

No Comments

Additional comments

In this paper, the authors try to discuss the effect of soil biota groups on the plant litter decomposition mediated by soil uses. The experimental design and investigation are rigorous, and the data sets are charming. However, the writing of the paper is too primary, and need to be revised before publication.
The discussion section should be rewrite carefully, and could separate from the result and discussion section. In my opinion, it is important to discuss the mechanism on how the soil uses influenced the soil biota groups and then influenced the plant litter decomposition in this section. Also, a robust discussion on the hypothesis is needed in this section. In addition, the conclusion should be shortened, and some contents in this section could be moved to discussion section or be deleted.

Reviewer 3 ·

Basic reporting

About the structure of the manuscript: Conclusions section is too long. Large part of its contents should be moved to the Results & Discussion section.
Specific comments on the text:
Line 14-15: "..and placed on the same soil.. different use intensities" insert under.
Lines 43-48: Sentence unclear. "The activity of macroorganisms is particularly important ..... in areas where climate is more constant ..". (?) Please re-write it.
All Figures: Part of the figure's captions includes the same text. I suggest referring the explanations to Figure 1.

Experimental design

No comments

Validity of the findings

No comments

Additional comments

The manuscript presents interesting results on the role of soil faunal groups on litter decomposition. I suggest highly reducing the conclusion section and moving part of the text to Results & Discussion, In this way, the manuscript would be clearer and the text less redundant.

---

## Round 0.2 · accepted · Accept

The paper has been improved by the authors substantially, and is now acceptable for publication in principle. A final language check could add to the quality of the paper as a whole, so, please do so.